# Study on the Detection of Defoliation Effect of an Improved YOLOv5x Cotton

**Xingwang Wang** [1,2], **Xufeng Wang** [1,2], **Can Hu** [1,2,*] , **Fei Dai** [2,3], **Jianfei Xing** [1,2], **Enyuan Wang** [1,2], **Zhenhao Du** [1,2], **Long Wang** [1,2] and **Wensong Guo** [1,2]

1 College of Mechanical and Electrical Engineering, Tarim University, Alar 843300, China
2 Modern Agricultural Engineering Key Laboratory at Universities of Education Department of Xinjiang Uygur Autonomous Region, Tarim University, Alar 843300, China
3 College of Mechanical and Electrical Engineering, Gansu Agricultural University, Lanzhou 730070, China
* Correspondence: 120140004@taru.edu.cn

**Abstract:** In order to study the detection effect of cotton boll opening after spraying defoliant, and to solve the problem of low efficiency of traditional manual detection methods for the use effect of cotton defoliant, this study proposed a cotton detection method improved YOLOv5x+ algorithm. Convolution Attention Module (CBAM) was embedded after Conv to enhance the network's feature extraction ability, suppress background information interference, and enable the network to focus better on cotton targets in the detection process. At the same time, the depth separable convolution (DWConv) was used to replace the ordinary convolution (Conv) in the YOLOv5x model, reducing the convolution kernel parameters in the algorithm, reducing the amount of calculation, and improving the detection speed of the algorithm. Finally, the detection layer was added to make the algorithm have higher accuracy in detecting small size cotton. The test results show that the accuracy rate P (%), recall rate R (%), and mAP value (%) of the improved algorithm reach 90.95, 89.16, and 78.47 respectively, which are 8.58, 8.84, and 5.15 higher than YOLOv5x algorithm respectively, and the convergence speed is faster, the error is smaller, and the resolution of cotton background and small target cotton is improved, which can meet the detection of cotton boll opening effect after spraying defoliant.

**Keywords:** defoliating agents; cotton detection; deep learning; defoliation effect

## 1. Introduction

Cotton was one of the most important crops in the world. It is also an important industrial raw material and strategic material. It plays an important role in the economic development of China and the world [1]. As the early cotton harvesting mode occupied a large number of human resources, the production cost per unit area of cotton was relatively high, which restricted the market competitiveness [2]. In recent years, with the rapid development and promotion of mechanized harvesting technology of cotton, the advantages of large-scale, standardized, and mechanized production of machine-picked cotton are prominent, and the economic benefits are very significant [3]. Chemical defoliation and ripening are important links of mechanical cotton picking and an important measure to promote the maturity of late maturing cotton [4]. The defoliating and ripening effect of the cotton defoliant can not only promote the natural cracking of cotton bolls and concentrated boll opening but also effectively reduce the impurity content of mechanically harvested cotton, which directly affects the quality and efficiency of mechanically harvested cotton [5,6]. Therefore, timely detection of the use effect of cotton defoliants and providing data support for the subsequent harvest of machine-picked cotton are the most important. The main basis of detection is the cotton ball opening rate. The traditional manual detection method not only requires a large number of laborers but also the results largely depend on the subjective perception of inspectors, which makes timely detection of cotton inefficient. Therefore, it is very necessary to improve cotton detection technology [7]. In recent years, deep learning has become an

increasingly popular method to solve problems in image classification, speech recognition, and video analysis [8]. Applications in the agricultural field include crop and organ classification, pest identification, fruit identification and counting, plant identification, plastic film cover identification, weed identification, and plant leaf analysis [9–15]. Most studies used convolutional neural networks, such as ResNet, VGG16, YOLO, and Faster R-CNN. Peng Mingxia et al., fused the feature pyramid network in the Faster R-CNN recognition algorithm to improve the image feature extraction ability of the algorithm and identify weeds in complex cotton fields; Yao Qing et al., adopted RetinaNet based on the resnext101 feature extraction network as the rice canopy pest detection algorithm and improved the recognition accuracy of the recognition algorithm by improving the feature extraction network in the network; Li Tianhua et al., Segmented the red area of Tomato by HSV method in the detection frame of YOLOv4 network and took the tomato whose segmentation area reached a certain proportion in the detection frame as the target output [16–18].

YOLO is a fast target detection algorithm using a convolutional neural network, which has great advantages in real-time detection applications [19]. The deep learning algorithm based on YOLO has been widely used in cotton detection and recognition. Wu Mingxiu et al. developed an improved YOLOv3 cotton heterosexual fiber detection method for cotton heterosexual fiber detection. Gu Wei et al. realized the detection of double-sided damage information and position information of population cotton seeds using YOLOv4. He Siqi et al. improved the YOLOv3 algorithm and applied it to the identification of cotton main stem growth points. Ouyang Yingjie et al. improved the YOLOv4 algorithm and deployed it on the cotton topping machine to identify the cotton terminal bud [20–23]. These cases show that the YOLO algorithm has great application significance in cotton detection.

In this study, the cotton sprayed with defoliant in the Alar City of Xinjiang was taken as the research object. In order to detect the effect of defoliant on the opening cotton boll, a small target improved YOLOv5x+ Cotton recognition method was proposed. The feature pyramid network and the attention (CBAM) mechanism are adopted to enable the algorithm to independently learn the weight of each channel [24,25], enhance the ability of the algorithm to distinguish cotton boll, and suppress the interference of background information. After the convolution attention module is embedded, the increase of algorithm parameters reduces the detection speed of the algorithm. Therefore, the deep separable convolution neural network is used instead of the ordinary convolution neural network in the original YOLOv5x to reduce the number of algorithm parameters and improve the detection speed of the algorithm [26]. At the same time, the output of detection results from the detection layer is increased. Combined with the fusion features of CSPNet, the feature fusion speed is optimized, and the accuracy of small target positioning and recognition is improved [27]. Finally, the accurate detection of the cotton boll opening effect after spraying defoliant is realized, in order to provide data support for cotton mechanical harvesting.

## 2. Materials Acquisition

### 2.1. Experiment Device and Materials

The test site was located in No. 3 Farm, Alar City, Xinjiang, China (40.541914° N, 81.285884° E). The cotton variety was Tahe No. 2, which was sown on April 25 and sprayed with defoliant on 5 September. The defoliant type was detalon 300 g/hm$^2$ + 40% ethephon 1200 mL/hm$^2$. The image acquisition time was from 15 September 2021 to 2 October 2021. This period was the most significant period for cotton defoliant and boll opening after spraying the defoliant. The image acquisition equipment was DJI Mavic AIR 2 four-axis UAV. The main parameters of the aircraft were the weight of 0.57 Kg, the maximum ascending speed of 14.4 Km/h, the maximum descending speed of 10.8 Km/h, the flying speed of 68.4 Km/h, the maximum flying time of 34 min, and the endurance mileage of 18.5 Km. The built-in camera had a wide-angle fixed focus lens, a coverage angle of 84 degrees, a sensor model of 1/2 inch CMOS, an effective pixel of 12 million, and a maximum photo size of 8000 × 6000 pixels. During shooting, the flying height was set to

5 m, the wide angle of the camera is 84°, and the shooting method is shown in Figure 1. A total of 1542 effective pictures of cotton containing defoliant were obtained.

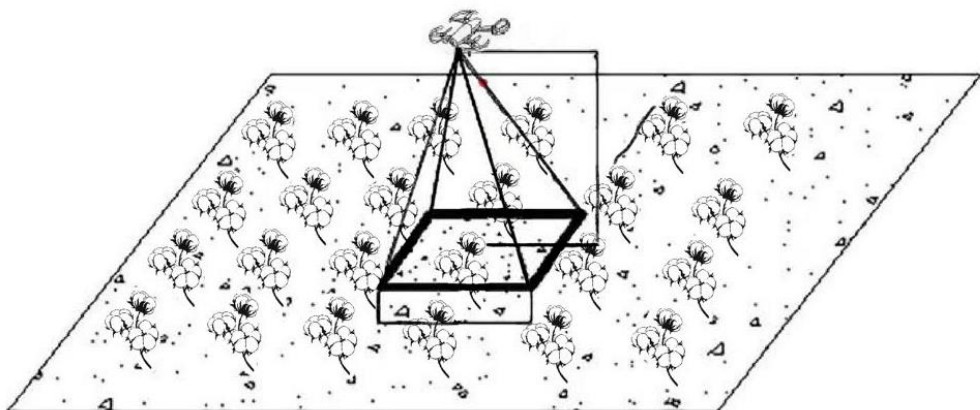

**Figure 1.** Shooting method.

In order to further increase the diversity of samples, properly control the number of samples, and reduce the training time of the algorithm, 500 images were selected for data enhancement through five methods: rotation, brightening, increasing contrast, coloring and darkening, as shown in Figures 2 and 3. All unclear pictures after enhancement were removed with 2432 pictures retained.

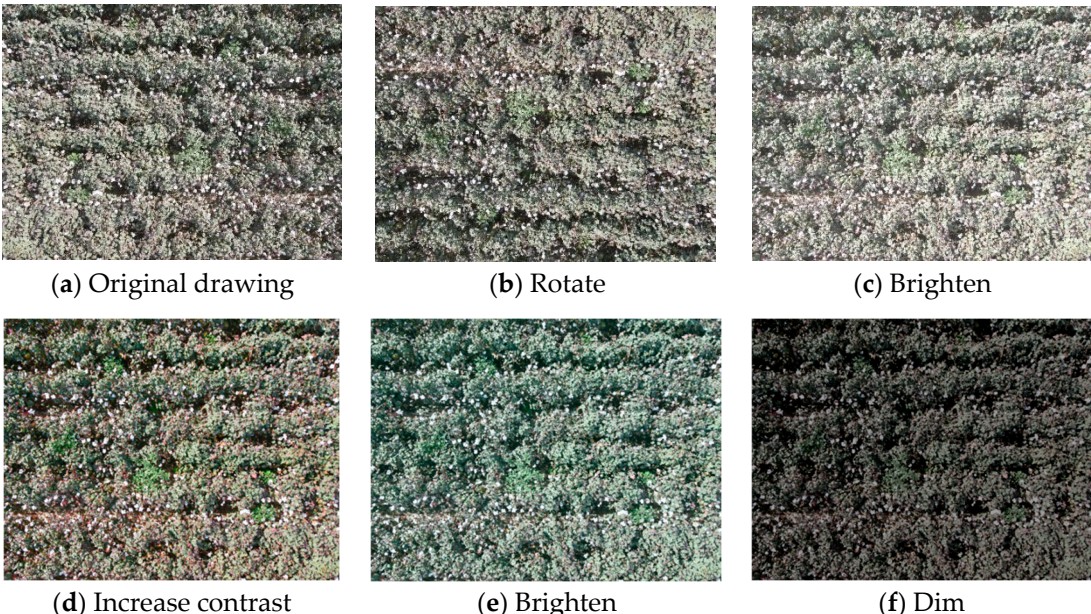

(**a**) Original drawing      (**b**) Rotate      (**c**) Brighten

(**d**) Increase contrast      (**e**) Brighten      (**f**) Dim

**Figure 2.** Original image and enhanced image.

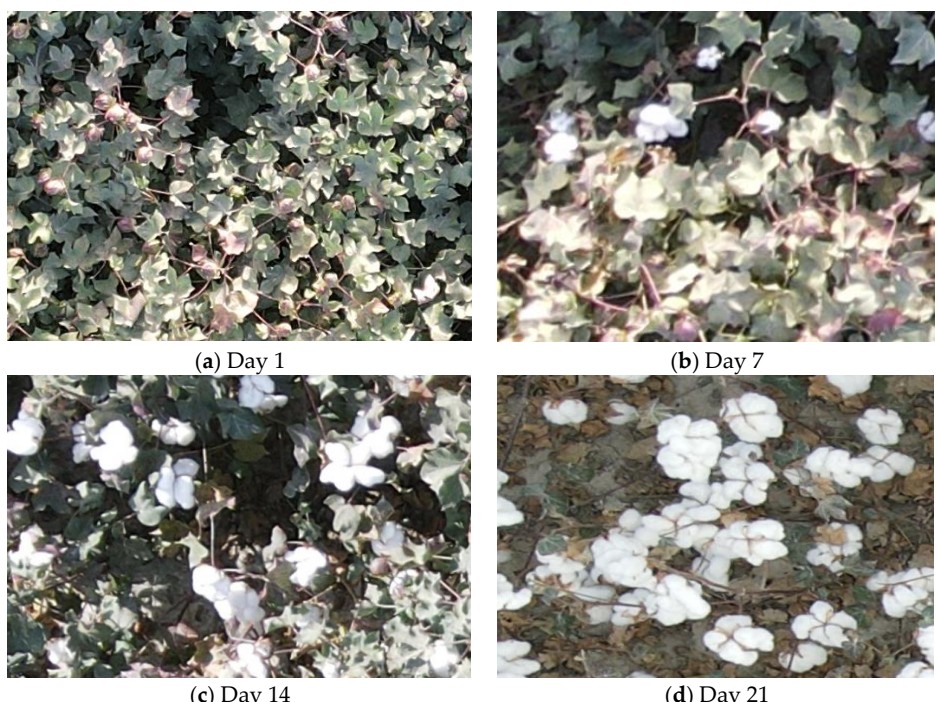

(**a**) Day 1

(**b**) Day 7

(**c**) Day 14

(**d**) Day 21

**Figure 3.** Images of cotton in different time periods after spraying defoliant.

### 2.2. Image Marking

Before the training, the cotton image was marked with the genie marking assistant software, the label data was added, and the export format was Pascal VOC format. Finally, each marked image generated a text file with a suffix of XML, which was used for the supervised learning of the deep learning algorithm. The marking format is shown in Figure 4.

The marked pictures were divided into training set, verification set, and test set according to the ratio of 8:1:1. Pascal voc2012 data set was used as the sample database format. The training set was used to train the algorithm parameters. The verification set tested the trained algorithm parameters and optimized the algorithm parameters according to the test results. Finally, the optimal recognition algorithm was selected. The test set was used to test the generalization ability of the training algorithm to recognize cotton.

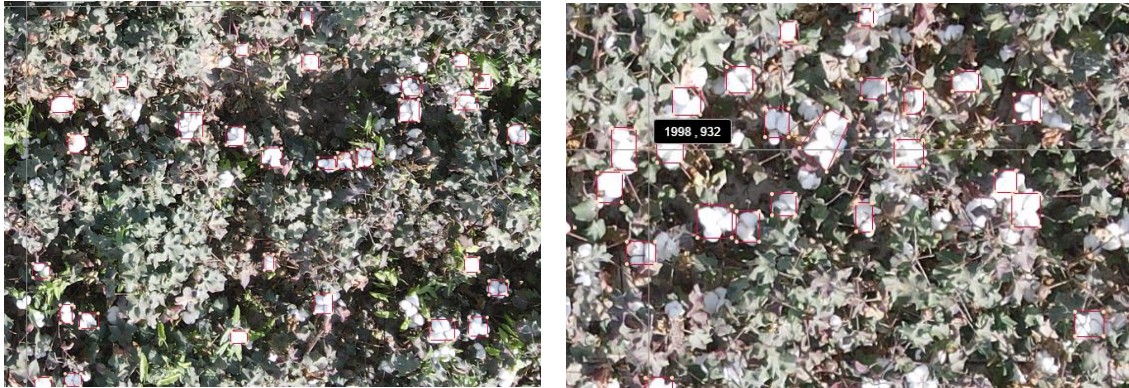

**Figure 4.** Image annotation.

### 3. Test Method

### 3.1. Detection Method Based on YOLOv5x

YOLO was widely used in the field of deep learning target recognition because it can see the whole image information and high processing speed during training and testing and

can make good use of the image information when detecting objects. The YOLO series algorithms have gone through four stages: YOLOv1, YOLOv2, YOLOv3, and YOLOv4, but the detection effect on small targets, adjacent near targets, and group targets is poor [28–30]. In order to adapt to more complex target detection, the Ultralytics team developed a new YOLOv5 algorithm in June 2020. YOLOv5 is a high-performance and universal target detection algorithm that can complete two tasks of target positioning and target classification at one time. The algorithm realizes the systematization of network architecture, namely YOLOv5s, YOLOv5m, YOLOv5l and YOLOv5x.The structures of these models are similar, but there are still some differences. As shown in Table 1, YOLOv5s uses one residual component in CSP, while in YOLOV5m, the network depth is increased, and in CSP, two residual components are used. In YOLOv5l, three residual components are used at the same position, and in YOLOv5x, four residual components are used. The number of convolution cores of the four YOLOv5 structures is also different, so it also directly affects the third dimension of the feature map after convolution. Taking the YOLOv5s structure as an example, in the Focus structure, the number of convolution cores during the final convolution operation is 32. Therefore, after the Focus structure, the size of the feature map becomes 30,430,432. The convolution operation in the Focus structure of YOLOv5m uses 48 convolution cores, so the feature map after the Focus structure becomes 30,430,448. The same principle applies to YOLOv5l and YOLOv5x. In YOLOv5, the more convolution kernels are, the greater the thickness of the feature map is, and the stronger the learning ability of the network to extract features is. Therefore, this study selected the YOLOv5x model as the experimental basis. The network structure of YOLOv5x is shown in Figure 5, which is mainly composed of four parts, namely, input end, Backbone, Neck, and Head.

**Table 1.** Characteristics of four YOLOv5 models.

| | YOLOv5s | YOLOv5m | YOLOv5l | YOLOv5x |
|---|---|---|---|---|
| CSP | 1 | 2 | 3 | 4 |
| Number of convolutional kernels | 32 | 48 | 64 | 80 |
| Feature map size | 30,430,432 | 30,430,448 | 30,430,464 | 30,430,480 |

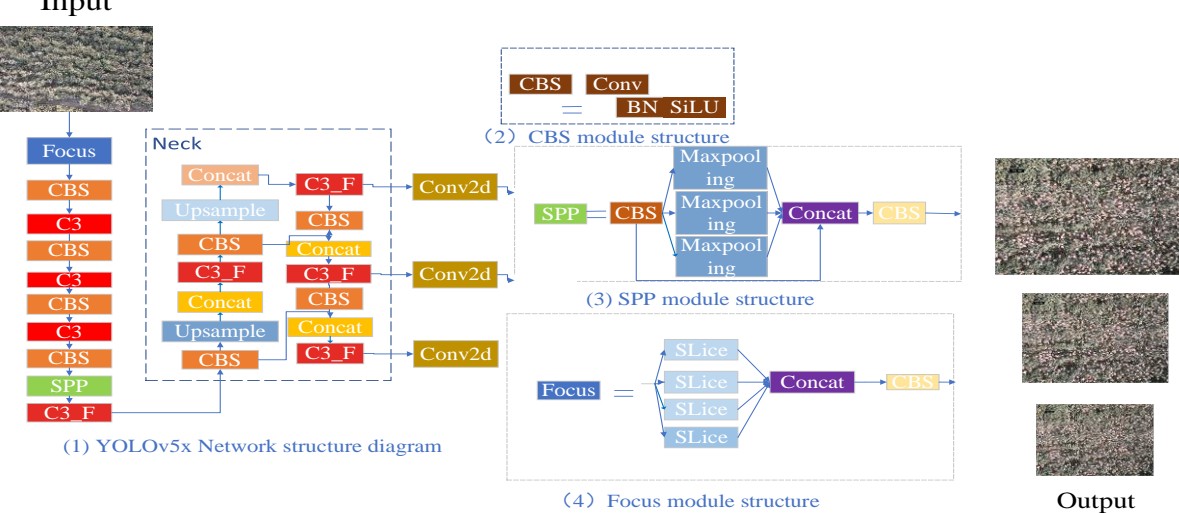

**Figure 5.** YOLOv5x network structure diagram.

The input terminal mainly includes three parts: automatic filling of pictures, data enhancement of pictures and anchor box calculation. The automatic filling of the picture adjusts the size of the original feature picture and unifies the input to the standard size; Mosaic technology is used for image data enhancement. Mosaic splices 4 feature images in a random arrangement and random clipping to enrich the background of feature images.

Anchor frame calculation calculates the size error between the predicted frame and the real frame output by the back propagation, and iteratively obtains the anchor frame with the most appropriate size.

Backbone is the feature extraction part of the network, mainly including Focus, CBS, and SPP modules. The Focus module slices the input feature map. The sliced feature map is stacked in channel dimensions. The size of the output feature map is 1/4 of the original and the number of channels is 4 times. After one conv (standard convolution), down sampling and feature extraction are realized. The principle is shown in Figure 5(4). The SPP module can convert an input feature map of any size into an output feature map of a fixed size and only needs to perform a standard convolution operation to reduce the amount of algorithm calculation. The CBS module is the main feature extraction module.

Feature fusion is carried out in the Neck part so that the feature map contains both detailed information and semantic information to improve the detection accuracy. The Head part outputs the category probability and frame position information of the target. This part is composed of three detection layers, which are used to detect targets of different sizes. Finally, the prediction frame and target category are marked in the image.

### 3.2. Detection Method Based on Improved YOLOv5x

#### 3.2.1. Attention Module (CBAM)

The convolutional attention module is a lightweight module, which is widely used in feedforward CNN. CBAM is composed of two parts: the channel attention module and the spatial attention module. In the channel attention module, feature mapping is input to the average pool layer and the maximum pool layer to obtain the average and maximum pool features respectively. These two features generate two sets of feature maps after sharing the DMLP composed of multiple perceptual layers and then superimpose them into one feature map after the Concat operation. After activating the function (ReLU module), the combined feature picture is output as a channel attention feature. The input amount and output amount of the channel attention module are multiplied, and the result is fed to the spatial attention module. The output amount of the CBAM module is obtained by multiplying the input amount and the output amount of the spatial attention module. In order to solve the problems of missing detection and false detection of small-sized cotton with a complex background such as uneven cotton boll opening and cotton leaf shading, CBAM is embedded after the common convolution Conv in the original YOLOv5x model to enhance the feature extraction ability of the network, so that the network can better focus on cotton targets and suppress the interference of background information in the detection process. The structure of CBAM is shown in Figure 6.

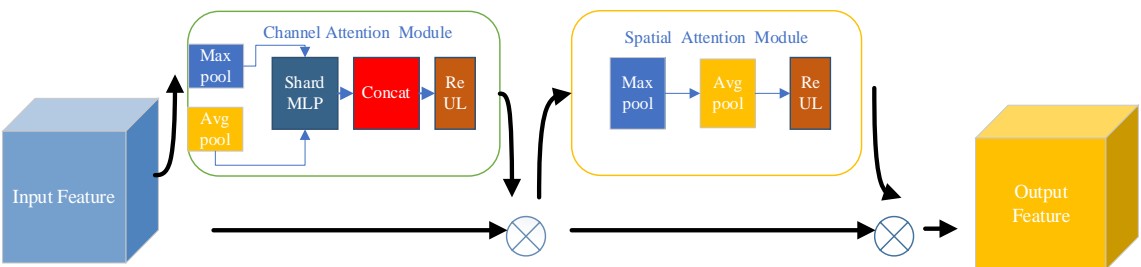

**Figure 6.** CBAM structure diagram.

#### 3.2.2. Deep Convolutional Neural Network

After embedding CBAM in the YOLOv5x algorithm, although the detection accuracy of the model is improved and the problems of missed detection and false detection are reduced, the parameter quantity of the model is also increased, and the detection speed of the model is reduced. In order to make the model have both high detection accuracy and fast detection speed, this study uses deep separable convolution (DWConv) to replace the

ordinary convolution (Conv) operation in the YOLOv5x model. The principle of the deep separable convolutional neural network is shown in Figure 6.

As shown in Figure 7, the depth-wise separable convolutional neural network is divided into two steps. Step 1, C convolution kernels of size K × K are used to perform channel by channel convolution on the input feature map, and the number of convolution kernels is the same as the number of channels on the previous layer (channel and convolution kernels correspond one to one), then C* H*W feature maps are obtained. Step 2 uses N filters (each filter consists of C 1*1 convolution kernels) Do a point-by-point convolution operation on the output feature map of step 1 respectively, and obtain the output feature map. Given an arbitrary input feature map $Fi \in RC*H*W$, to get the output feature map $FO \in RN*P*Q$, where C is the number of channels of the input feature map, H is the height of the input feature map, W is the width of the input feature map, the size of the convolution kernel is K*K, N is the number of channels of the output feature map, P is the height of the output feature map, Q is the width of the output feature map. If a deep separable convolutional neural network is used, its convolution kernel parameters are C*K*K + C*N*1*1, while using a common convolutional neural network, its convolution kernel parameter is C*K*K*C*N. Therefore, the use of depth-wise separable convolutional neural networks can reduce the amount of convolution kernel parameters in the algorithm, thereby reducing the amount of calculation and improving the detection speed of the algorithm.

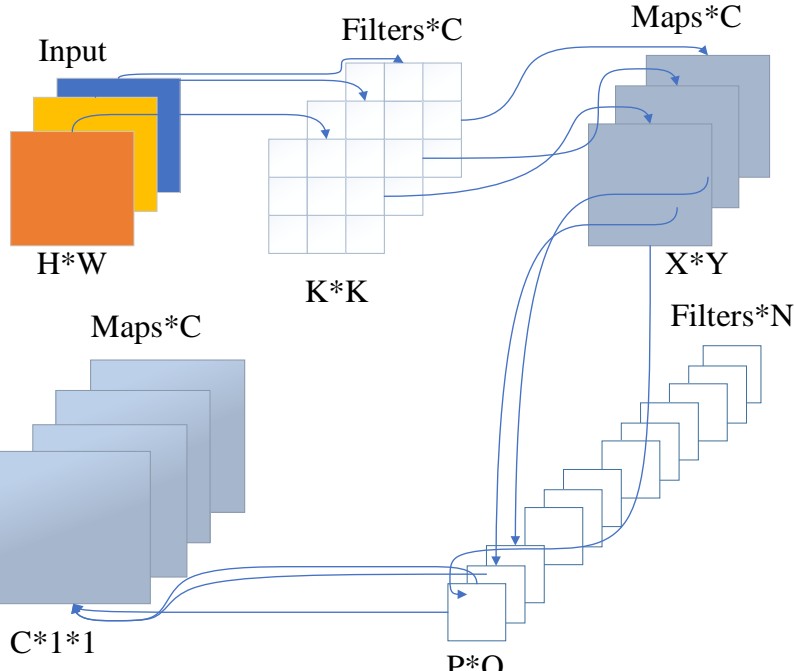

**Figure 7.** Deep separable convolutional neural network. (* Represents a multiple relationship).

### 3.3. Small Size Cotton Detection Layer

In the process of photographing cotton, due to the influence of flying height, the single area of cotton in the picture is too small. At the same time, it is also affected by problems such as cotton leaf color confusion and cotton leaf shading. In the YOLOv5x algorithm, the bottom feature map extracted by convolution kernel has a high resolution, clear color and texture, and accurate target position. However, with the deepening of the network, after multiple convolution operations, the semantic information of the extracted feature map is gradually enriched and contains more concept-level information, which is easier for a human to understand. However, the resolution of the picture is low, and the target position is rough. With the gradual deepening of the network, small-scale cotton targets may even be lost, resulting in missed detection and false detection.

Therefore, this paper proposes the YOLOv5x+ detection model which increases the small target detection layer to improve the accuracy of small target detection. In the original YOLOv5x model, the feature maps output from layers 17, 20, and 23 were used for small, medium, and large target detection, respectively, but because the small size cotton occupies a very small area in the data picture, the targets will be lost after multiple convolutions, resulting in missed detections. In order to improve the detection accuracy of small-sized cotton boll and solve the problems of the low resolution of the high-level feature map, rough target position, and even loss, on the basis of the original algorithm, the detection layer is added and replaced. After the 17th layer, the feature map is up-sampled, so that the feature map continues to expand. At the same time, at the 20th layer, the feature map with a size of $160 \times 160$ is concatenated with the feature map of the second layer in the backbone network, so as to obtain a larger feature map for small target detection. In the 31st layer, that is, the detection layer, a small target detection layer is added. A total of four layers [21,24,27,30] are used for detection. At the same time, C3 is replaced with bottleneckCSP in the backbone, so that the algorithm has higher accuracy in detecting small-sized cotton. The improved network diagram is shown in Figure 8.

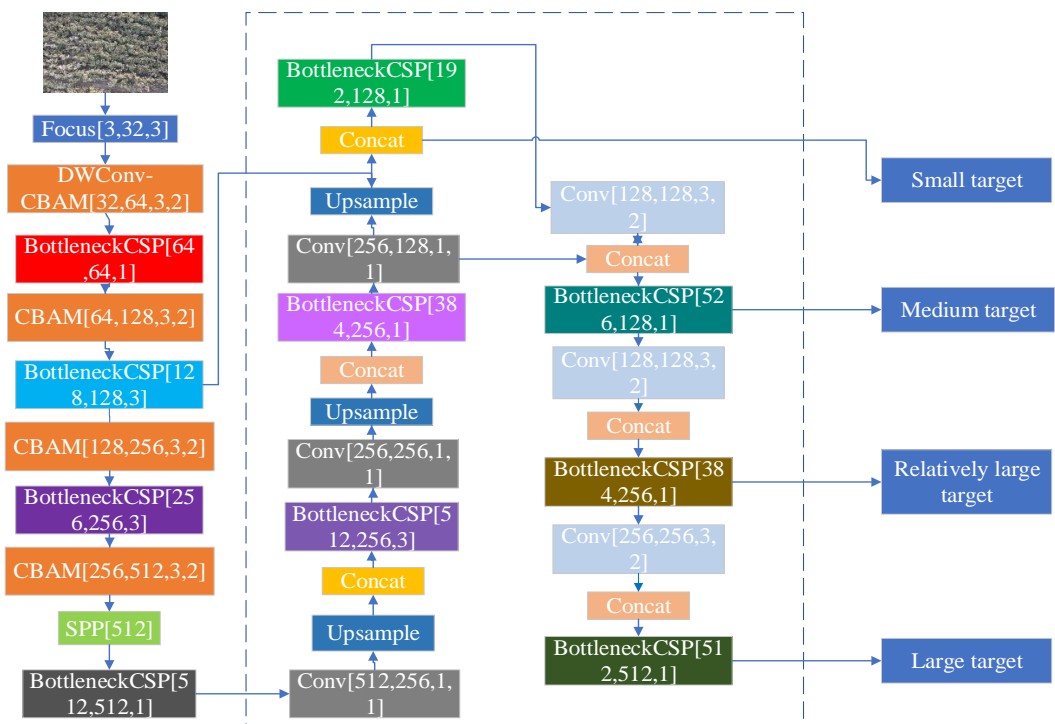

**Figure 8.** Improved YOLOv5x network structure diagram.

After the detection layer is added to YOLOv5x+, the problem is that the amount of calculation increases, the detection time increases, and the inference detection speed decreases. However, for small targets, there are indeed very good improvements.

## 4. Results

### 4.1. Training Platform

All training and testing methods of this test were conducted on a desktop Dell computer. The computer was configured with Intel (R) Core (TM) i9-10900K CPU, 3.70GHz main frequency, 4.00GB memory, 11GB GeForce RTX2080Ti graphics card, 800G hard disk, and the computer operating system was windows10 professional 64-bit Z490UD(DirectX12); The Darknet framework was used to build a network and attempts to use the PyTorch deep learning framework were made. The development environment was PyTorch 1.6.0, CUDA 11.0, CUDNN8.0.1 and python3.9.

*4.2. Network Training and Detection*

4.2.1. Algorithm Training Parameters

The model had 400 iterations (Epoch). The input image size was 640 × 640, and the batch size value was 32. Due to the learning rate being too large, the network may not converge. If the learning rate is too small, the network will converge too slowly. Therefore, the initial learning rate was set to 0.001 in this test. In order to prevent overfitting, the weight attenuation coefficient was set to 0.001, the initial confidence threshold was set to 0.1, the non-maximum inhibition threshold was set to 0.3, the learning rate momentum factor was 0.937, and the loss weight was set to λbox = 0.05; λconf = 1; λcls = 0.5; λobj = 1 and λnoobj = 0.5.

4.2.2. Algorithm Training Parameters

In this study, mean average precision (map) and algorithm detection speed were used as the evaluation indicators of the algorithm. mAP ∈ [0,1], the larger the value, the better the detection effect of the algorithm. The detection speed of the algorithm was obtained from the detection time of a single cotton image in the GPU environment of this study. The calculation equation of map is as follows (1) and (3).

$$P = \frac{TP}{TP + FP} \tag{1}$$

$$R = \frac{TP}{TP + FN} \tag{2}$$

$$mAP = \frac{\sum_{k=1}^{N} PR}{N} \tag{3}$$

where: P is the detection accuracy, that is, the ratio of the correctly detected cotton boll opening to the total number of detected cotton. R is the recall rate, that is, the ratio of the number of correctly detected cotton to all detected objects in the data set. *TP* is the number of true samples, that is, the number of cotton that has been correctly detected, *FP* is the number of false positive samples, that is, the number of cotton that has been erroneously detected. *FN* is the number of false negative samples, i.e., the number of non-cotton samples, such as cotton leaves, etc., and *N* is the number of species in the sample.

At the same time, according to the number of cotton in the whole image selected by the detection frame, calculate the proportion of the area identified in cotton. The higher the proportion, the greater the probability that the detection result is cotton, but the number of missed cotton will also increase; The lower the proportion, the less the number of missed cotton. The total number of target cotton and the number of unrecognized cotton are taken as the total (Sum, *S*), the proportion of identified cotton in the total is taken as the recognition accuracy rate (Accuracy *A*, %), the proportion of unrecognized non-cotton is the error rate (Error rate *E*, %), and the proportion of undetected cotton is the missing recognition rate (Missing recognition rate, *M*, %). The calculation equation is (4) and (6):

$$A = \frac{N_1}{S} \times 100\% \tag{4}$$

$$E = \frac{N_2}{S} \times 100\% \tag{5}$$

$$M = \frac{N_3}{S} \times 100\% \tag{6}$$

where *A* is the correct rate, %; *E* is the error rate, %; *M* is the missed detection rate, %; *S* is the total number of cotton; *N1* is the number of identified cotton; *N2* is the number of misidentified non cotton; *N3* is the number of unrecognized cotton.

### 4.2.3. Determination of Optimal Threshold

The recognition model needed to filter the prediction box according to the preset threshold after predicting the confidence level of the target category. Based on the same recognition model, different confidence thresholds were used for prediction, and the accuracy and recall of the recognition results were different.

If the confidence threshold of the recognition model was not properly selected, the prediction results shown in Figure 9 would appear: when the confidence threshold was set too low, cotton leaves in the image would be identified by mistake, and when the threshold was set too high, cotton targets may be missed. Therefore, it was necessary to determine the appropriate confidence threshold for the model in combination with specific recognition tasks to accurately screen the cotton targets to be identified.

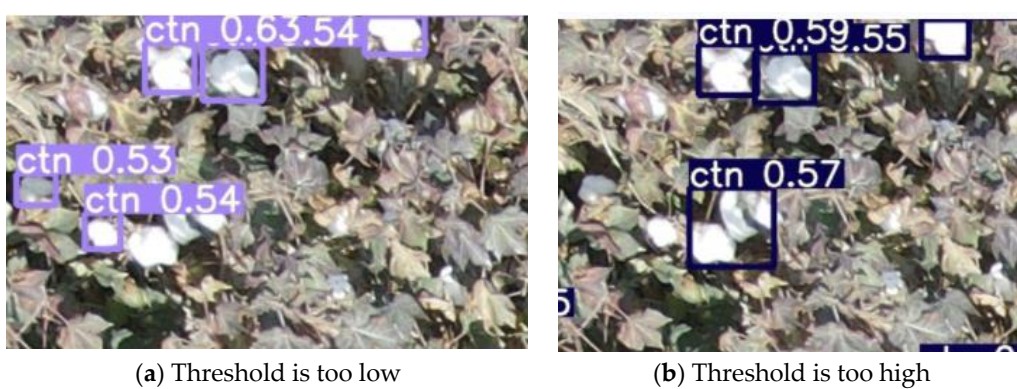

(**a**) Threshold is too low        (**b**) Threshold is too high

**Figure 9.** Impact of Threshold on Recognition Results.

Based on the cotton recognition model obtained from the training, by adjusting the confidence threshold, the changes in the accuracy, recall, and mAP of the measurement model for cotton target recognition of the test set under different thresholds were compared, and the best prediction category threshold of the model in combination with the actual needs of cotton recognition tasks was determined. From this test, the identification accuracy, recall, and mAP distribution of the model under different confidence thresholds are shown in Figure 10.

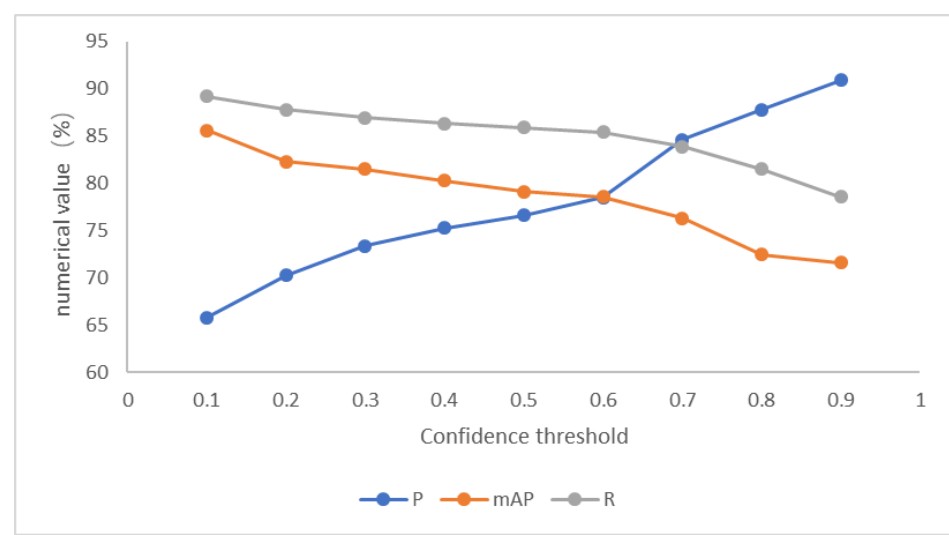

**Figure 10.** Changes of performance parameters of model with different confidence thresholds.

*4.3. Analysis of Test Results*

In order to verify the effect of the improved model, the indexes of the improved model were compared with the original YOLOv5x model, and the results are shown in Figure 11.

It can be seen from Figure 11a that after 100 iterations, the loss curve of the improved model tended to be stable, and the loss rate was lower than that of the original algorithm, indicating that the improved model converged faster and the loss value was smaller. The initial loss value of the first 20 iterations of the confidence loss was less than 0.15, and converged rapidly, indicating that the improved model had higher discrimination ability between cotton background and cotton, higher accuracy in detecting cotton boll opening and missed detection, therefore, the problem of false detection was solved. It can be seen from Figure 11b that the map (%) value of the improved algorithm was 78.47, an increase of 5.15 compared with 73.32 of the original algorithm. Due to the increase of the detection layer, the initial value of the map increased slowly, and the initial fluctuation of the iteration curve of the improved algorithm was large. It tended to be stable after 290 iterations.

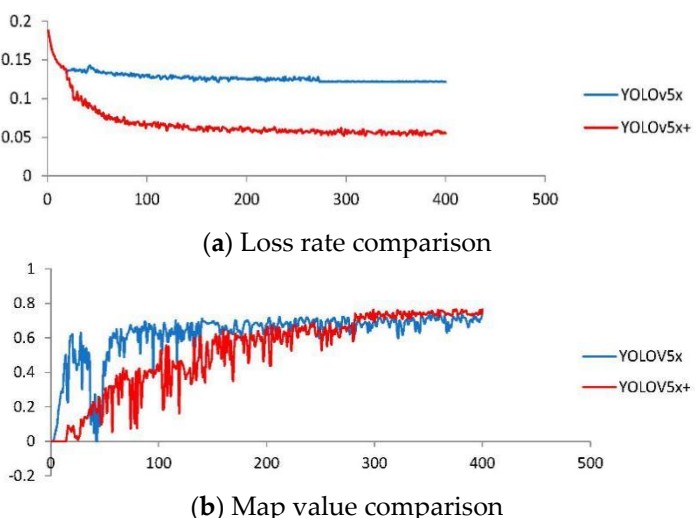

(**a**) Loss rate comparison

(**b**) Map value comparison

**Figure 11.** Comparison of test results.

*4.4. Ablation Test*

The improved YOLOv5x+ model embedded a CBAM module, replaced Conv with DWConv, and added a detection layer for small-size cotton. In order to verify the optimization effect of each improvement on the original model, the ablation test was designed. In order to ensure the correctness of the ablation test, the super parameter settings and operating environment of the model were the same. The results are shown in Table 2.

**Table 2.** Ablation test results.

| Model | CBAM | DWConv | Small Size Cotton Detection Layer | P/% | R/% | Map/% | Parameter Quantity | Detection Time/ms |
|---|---|---|---|---|---|---|---|---|
| YOLOv5x | × | × | × | 82.37 | 80.32 | 73.32 | 6 187 024 | 43.78 |
| A | √ | × | × | 84.59 | 82.96 | 72.43 | 4 082 034 | 49.21 |
| B | × | √ | × | 84.13 | 85.41 | 73.54 | 6 820 835 | 43.12 |
| C | × | × | √ | 86.64 | 84.40 | 75.19 | 6 759 216 | 65.76 |
| YOLOv5x+ | √ | √ | √ | 90.95 | 89.16 | 78.47 | 5 626 486 | 63.43 |

It can be seen from the results in Table 2 that after the CBAM module was added to model A, the parameters of the original model were reduced by 2,104,990, and the detection time was 43.78 ms. Compared with the original YOOv5x model, the time used to detect a single sample image was increased by 5.43 ms. The model detection speed

was reduced, the accuracy rate P (%) was 84.59, which was 2.22 more than the original YOOv5x model, and the recall rate R (%) was 82.94, which was 2.74 more than the original YOOv5x model. The mAP value decreased slightly. Due to DWConv being used in model B, the feature extraction ability was improved. Compared with the original model, the parameter quantity of model b was increased by 633,811, the detection time was decreased by 0.66 ms, the accuracy rate P (%) was increased by 1.76, the recall rate R (%) was increased by 5.09, and the mAP value was increased by 0.22. The detection layer for small-size cotton was added to model C. Therefore, the detection time was increased. Compared with the original YOLOv5x model, the detection time was increased by 21.98 ms, the parameter quantity was increased by 572,192, the accuracy rate P (%) was increased by 4.27, the recall rate R (%) was increased by 4.06, and the mAP was increased by 1.87.

The accuracy P (%), recall R (%), and map values of the improved YOLOv5x+ model were 90.95, 89.16, and 78.47, and the performance was further improved. Compared with the original YOLOv5x model, the accuracy rate P (%), the recall rate R (%), and the map were increased by 8.58, 8.84, and 5.15 respectively. Although the detection time of a single image was 19.65 ms longer than that of the original model, it met the needs of cotton detection tasks and was more suitable for cotton detection tasks after spraying defoliant.

### 4.5. Comparison of Different Network Model Training

In this study, ResNet-50, ResNet-18, and DesNet-201 networks were used to compare the performance of the improved models.

Table 3 shows that YOLOv5x+ had the highest mAP value and the highest detection accuracy. Compared with ResNet-50, YOLOv5x+'s mAP was 7.34% higher. Compared with ResNet-18 and DesNet-201 models, the mAP of YOLOv5x+ increased by 7.89% and 2.89%, respectively. In addition, the model proposed in this paper had significantly improved precision, recall, and detection time. Based on the comparison of the above experimental data, the improvement of this study is meaningful.

**Table 3.** Performance comparison of different models.

|  | P/% | R/% | Map/% | Detection Time/ms |
|---|---|---|---|---|
| ResNet-50 | 80.10 | 71.67 | 71.14 | 74.86 |
| ResNet-18 | 79.13 | 67.14 | 70.58 | 80.16 |
| DesNet-201 | 84.30 | 74.95 | 75.58 | 69.73 |
| YOLOv5x+ | 90.95 | 89.16 | 78.47 | 63.43 |

### 4.6. Analysis of Test Results

In order to visually verify the detection effect, the same data was selected for the two models for detection, and the detection results are shown in Figure 12. It can be seen that the accuracy of the YOLOv5x+ model detection was significantly higher than that of the original YOLOv5x model. The detection accuracy has been improved by more than 20%. It could more accurately identify small target cotton, reduce false detection and missed detection, and it improved the detection effect significantly.

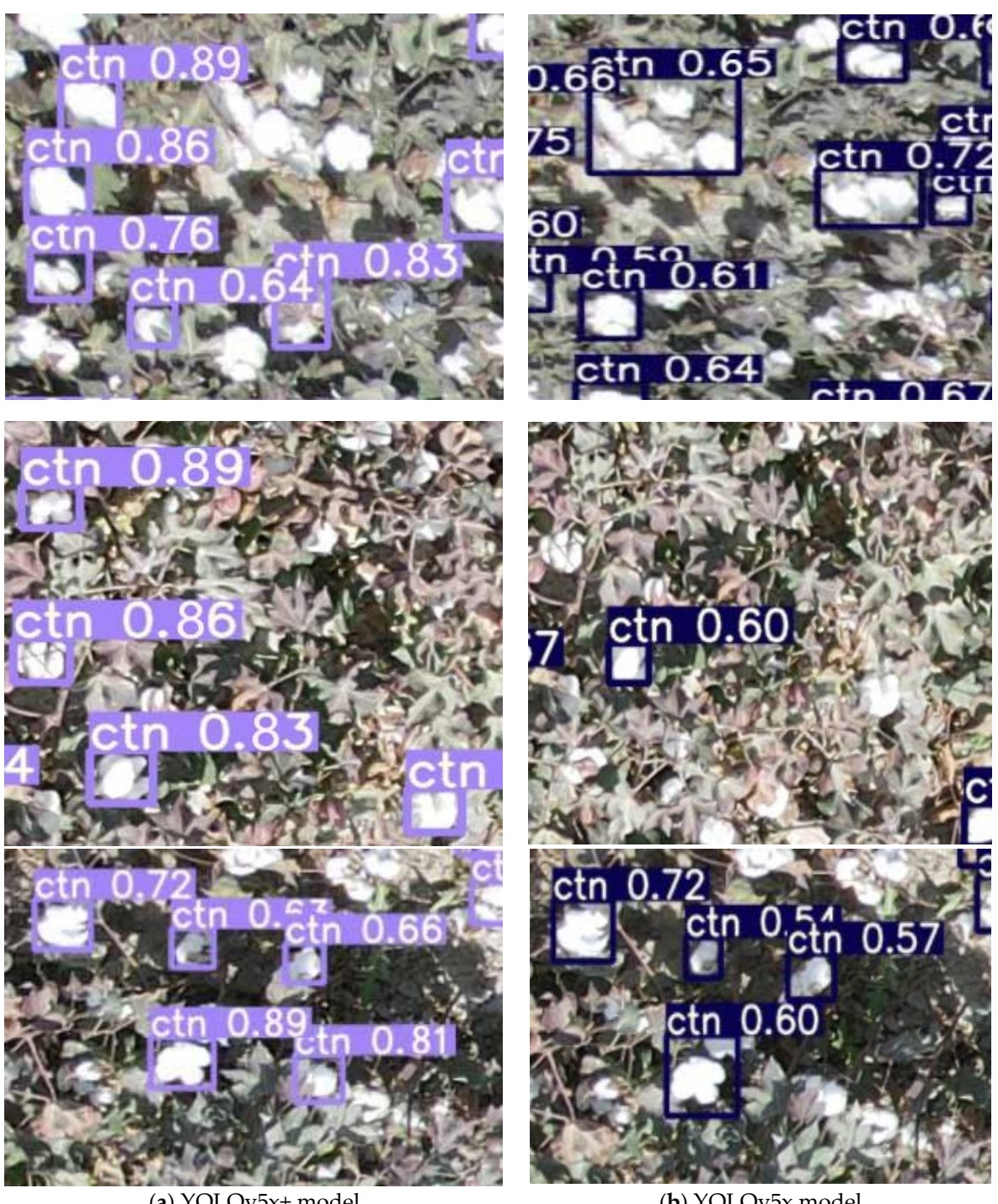

(**a**) YOLOv5x+ model            (**b**) YOLOv5x model

**Figure 12.** Comparison of test results.

To summarize, the improved YOLOv5x+ target detection model has advantages in accuracy and missed detection rate in comparison with the YOLOv5x model. It can accurately identify and locate the cotton after spraying defoliant. Although the detection time of a single image is increased, it is more suitable for cotton detection tasks than the original model.

## 5. Conclusions

In this study, given the low efficiency of traditional cotton defoliant use effect detection and the low detection effect of cotton boll opening effect, based on the original YOLOv5x

model, a depth learning model YOLOv5x+ for detecting the effect of cotton defoliant was designed by embedding convolution attention module, using depth separable convolution, and adding detection layers for small size cotton. It can be seen from the test results that the average detection time of each image of the YOLOv5x model is 78.43ms, and the accuracy rate P (%) and recall rate R (%) are 90.95 and 89.16, which are 8.58 and 8.84 percentage points higher than the original YOLOv5x model, respectively, meeting the cotton detection task and having higher accuracy, so it applies to the cotton detection field. However, there are still cases of missing or error detection. The next part of the research will need to further optimize the attention mechanism of the model, improve the detection accuracy of the model, and deploy the improved model on the UAV to achieve real-time cotton detection.

**Author Contributions:** Writing—original draft preparation, X.W. (Xingwang Wang); writing—review and editing, C.H.; Resources, X.W. (Xufeng Wang); data curation, L.W.; visualization, J.X.; supervision, W.G.; investigation, E.W. and Z.D.; project administration, C.H. and F.D. All authors have read and agreed to the published version of the manuscript.

**Funding:** This work was supported by the science and technology planning project of the first division of alar city, grant number 2022XX06, the Open project of Key Laboratory of modern agricultural engineering, grant number TDNG2021101 and the Innovation research team project of Tarim University, grant number TDZKCX202103.

**Institutional Review Board Statement:** Not applicable.

**Informed Consent Statement:** Not applicable.

**Data Availability Statement:** The data presented in this study are available on request from the corresponding author.

**Acknowledgments:** Thanks for grateful to the science and technology planning project of the first division of alar city (2022XX06), the Open project of modern agricultural engineering (TDNG2021101) and the Innovation research team project of Tarim University, grant number TDZKCX202103. The authors are grateful to anonymous reviewers for their comments.

**Conflicts of Interest:** The authors declare no conflict of interest.

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
