# Peer review of "Study on the Detection of Defoliation Effect of an Improved YOLOv5x Cotton"

_agriculture, doi:10.3390/agriculture12101583_

Round 1
Reviewer 1 Report
The authors dealt with a very important and necessary topic in agricultural practice, including the cultivation of cotton. The improvement of the YOLOv5x + algorithm allows for a precise interpretation of the effectiveness of defoliants. The research undertaken and the improved model can be used in the cultivation of other species.
1. There is no information in the Materials acquisition chapter on the agrotechnical aspects of cotton cultivation. I propose to supplement: the name of the cotton variety and the type of defoliant, the date of its use.
2. No discussion, the authors provide some information in the introduction, but after presenting their own research results, they should be compared with the achievements so far.
Author Response
Dear reviewer,
Thank you very much for giving us an opportunity to revise our manuscript entitled "Study on the detection of defoliation effect of an improved YOLOv5x cotton"(Manuscript ID:agriculture-1926388). We have studied the comments carefully. Those comments are valuable and helpful for revising and improving our paper, and they provide important guiding significance to our researchers.
We have studied comments carefully and have made correction which we hope meet with approval. Revised portion are marked in red in the paper.. In addition to the suggestions and problems put forward by experts, we have also modified and identified other problems found during the review.
The main correction in the paper and the responds to reviewer’s comments are as following:
Reviewer: 1
Comments and Suggestions for Authors:
1.There is no information in the Materials acquisition chapter on the agrotechnical aspects of cotton cultivation. I propose to supplement: the name of the cotton variety and the type of defoliant, the date of its use.
.
Response: Thank you for your suggestions. It's our fault that we didn't provide information on agricultural technology of cotton planting. According to your suggestions. We have added the sentence “The cotton variety is Tahe No. 2, which was sown on April 25 and sprayed with defoliant on September 5. The defoliant type is detalon 300g/hm2+40% ethephon 1200mL/hm2. The image acquisition time is from September 15, 2021 to October 2, 2021. This period is the most significant period for cotton defoliant and boll opening after spraying the defoliant.”(Line 97-101) in the manuscript.
- No discussion, the authors provide some information in the introduction, but after presenting their own research results, they should be compared with the achievements so far.
Response: Thank you for your suggestions.Response: Thank you for your comment about details.Thank you for your guidance on our shortcomings. According to your suggestions, we added comparative experiments and discussed them. The results are as follows:
“4.5. Comparison of different network model training
In this study, ResNet-50, ResNet-18 and DesNet-201 networks are used to compare the performance of the improved models.
Table 3 shows that YOLOv5x+has the highest mAP value and the highest detection accuracy. Compared with ResNet-50, YOLOv5x+'s mAP is 7.34% higher. Compared with ResNet-18 and DesNet-201 models, the mAP of YOLOv5x+is increased by 7.89% and 2.89% respectively. In addition, the model proposed in this paper has significantly improved in precision, recall and detection time. Based on the comparison of the above experimental data, the improvement of this study is meaningful.
Table 3 Performance comparison of different models
|
P/% |
R/% |
Map/% |
Detection time/ms |
ResNet-50 |
80.10 |
71.67 |
71.14 |
74.86 |
ResNet-18 |
79.13 |
67.14 |
70.58 |
80.16 |
DesNet-201 |
84.30 |
74.95 |
75.58 |
69.73 |
YOLOv5x+ |
90.95 |
89.16 |
78.47 |
63.43 |
”(Line 416-425)

Reviewer 2 Report
In this work, a modified deep learning method is used to evaluate the effect of cotton defoliants, in which the convolution attention module and detection layers are used to improve the detection accuracy of small-size cotton. The experiments show that the classification indexes P-value and R-value are increased by 8.58% and 8.84%, respectively compared with the conventional methods. Overall, the topic is interesting for scholars who are studying cotton production. However, some concerns need to be addressed before acceptance for publication.
1. It is suggested to provide more details about the difference between YOLOv5s, YOLOv5m, YOLOv5l, and YOLOv5x.
2. A legend is suggested to add in Fig. 2 to show the detection objectives intuitively.
3. The text in Fig 4 and Fig 7 should be enlarged.
4. Fig. 8 is suggested to redraw using line-symbol.
5. The writings should be polished carefully. Some sentence is confusing and difficult to understand. For example, “uses C convolution kernels with a size of K*K to perform channel-by-channel convolution on the input feature map”.
7. It seems that the reference format is not the style required by the journal.
Author Response
Dear reviewer,
Thank you very much for giving us an opportunity to revise our manuscript entitled "Study on the detection of defoliation effect of an improved YOLOv5x cotton"(Manuscript ID:agriculture-1926388). We have studied the comments carefully. Those comments are valuable and helpful for revising and improving our paper, and they provide important guiding significance to our researchers.
We have studied comments carefully and have made correction which we hope meet with approval. Revised portion are marked in red in the paper.. In addition to the suggestions and problems put forward by experts, we have also modified and identified other problems found during the review.
The main correction in the paper and the responds to reviewer’s comments are as following:
Reviewer: 3
Comments and Suggestions for Authors:
- The abstract is not clear, please highlight the innovative points of the manuscript;
Response: Thanks for your careful review.Sorry for the unclear expression of our abstract.According to your suggestion, we have modified the abstract, which is as follows:
“In order to study the detection effect of cotton boll opening after spraying defoliant, and to solve the problem of low efficiency of traditional manual detection methods for the use effect of cotton defoliant, this study proposed a cotton detection method improved YOLOv5x+algorithm. Convolution Attention Module (CBAM) is embedded after Conv to enhance the network's feature extraction ability, suppress background information interference, and enable the network to better focus on cotton targets in the detection process; At the same time, the depth separable convolution (DWConv) is used to replace the ordinary convolution (Conv) in YOLOv5x model, reducing the convolution kernel parameters in the algorithm, reducing the amount of calculation, and improving the detection speed of the algorithm; Finally, the detection layer is added to make the algorithm have higher accuracy in detecting small size cotton. The test results show that the accuracy rate P (%), recall rate R (%) and mAP value (%) of the improved algorithm reach 90.95, 89.16 and 78.47 respectively, which are 8.58, 8.84 and 5.15 higher than YOLOv5x algorithm respectively, and the convergence speed is faster, the error is smaller, and the resolution of cotton background and small target cotton is improved, which can meet the detection of cotton boll opening effect after spraying defoliant.”(Line 15-29)
- In lines 74-75 and 245-247, English needs to be improved;
Response: Thanks for your careful review.We have revised our manuscript based on your comments.Substituting “These research cases show that YOLO algorithm has great potential in cotton detection.”(Line 76-77) with “These cases show that YOLO algorithm has great application significance in cotton detection.”(Line 76-77). Substituting “In the original YOLOv5x model, the feature maps output from the 17th, 20th and 23rd layers are used to detect small, medium and large targets respectively. However, due to the small area of small-sized cotton in the data picture, the target will be lost after multiple convolutions, resulting in missed detection.”(Line 261-265) with “In the original YOLOv5x model, the feature maps output from layers 17, 20, and 31 were used for small, medium, and large target detection, respectively, but because the small size cotton occupies a very small area in the data picture, the targets will be lost after multiple convolutions, resulting in missed detections. ”(Line 261-265)
- Figure 3 is not clear enough, and the marked details are not displayed enough;
Response: Thanks for your careful review.We apologize for the mistake in the text.Now we've made the following modifications to Figure 3
Fig. 3 Image annotation
- In line 229 of the manuscript, why is the feature map output at the 17th, 20th and 23rd layers processed, what is the idea? What is the size of the feature maps output by layers 17, 20 and 23?
Response: Thanks for your careful review.Why is the feature map output at the 17th, 20th and 23rd layers processed?Because the feature maps output from the 17th, 20th and 23rd layers are used to detect small, medium and large targets respectively, when the UAV takes a picture of cotton, due to the influence of flight height, the size of cotton in the picture is inconsistent, and the edge cotton and cotton covered by leaves account for a small proportion in the picture. In order to display cotton, the feature maps on the corresponding layers are processed.The output feature map sizes of layers 17, 20 and 23 are 80*80, 160*160, 40*40, respectively.
- Figure 7 is not pretty. Please use different color squares to represent different types of convolution and modules;
Response: Thanks for your suggestion.According to your suggestion, we used different colored squares to represent different types of convolutions and modules, and modified the text font size.
Fig. 7 Improved YOLOv5x network structure diagram
- In line 260-261, the classification threshold has a great influence on the experimental results. How is the threshold obtained? Why is there no analysis? Please add this part of the experiment to determine the optimal threshold;
Response: Thanks for your careful review.We are sorry for this mistake.According to your suggestion, we have added this part of experiment.The details are as follows:
“4.3.3. Determination of optimal threshold
The recognition model needs to filter the prediction box according to the preset threshold after predicting the confidence level of the target category. Based on the same recognition model, different confidence thresholds are used for prediction, and the accuracy and recall of the recognition results are different.
If the confidence threshold of the recognition model is not properly selected, the prediction results shown in Fig. 8 will appear: when the confidence threshold is set too low, cotton leaves in the image will be identified by mistake, and when the threshold is set too high, cotton targets may be missed. Therefore, it is necessary to determine appropriate confidence threshold for the model in combination with specific recognition tasks to accurately screen the cotton targets to be identified.
(a)Threshold is too low (b)Threshold is too high
Fig. 8 Impact of Threshold on Recognition Results
Based on the cotton recognition model obtained from the training, by adjusting the confidence threshold, compare the changes in the accuracy, recall and mAP of the measurement model for cotton target recognition of the test set under different thresholds, and determine the best prediction category threshold of the model in combination with the actual needs of cotton recognition tasks. Through test, the identification accuracy, recall and mAP distribution of the model under different confidence thresholds are shown in Fig. 9.
Fig.9 Changes of performance parameters of model with different confidence thresholds
”(Line 337-359)
- Several parameters that must be important in algorithm improvement,, such as model size, model parameters, model FLOPs, AUC index, etc., are not reflected in Table 1. And Table 1 shows that the detection time of the improved model is 50% higher than that of the original model. Is it worth it?
Response: Thanks for your careful review.The parameter problem in the algorithm improvement is not shown in Table 2, which is our fault. Thank you again for pointing out the shortcomings. Now we have added the parameter quantity index to the table, and the improvement is as follows:
Table 2 ablation test results
Model |
CBAM |
DWConv |
Small size cotton detection layer |
P/% |
R/% |
Map/% |
Parameter quantity |
Detection time/ms |
YOLOv5x |
× |
× |
× |
82.37 |
80.32 |
73.32 |
6 187 024 |
43.78 |
A |
√ |
× |
× |
84.59 |
82.96 |
72.43 |
4 082 034 |
49.21 |
B |
× |
√ |
× |
84.13 |
85.41 |
73.54 |
6 820 835 |
43.12 |
C |
× |
× |
√ |
86.64 |
84.40 |
75.19 |
6 759 216 |
65.76 |
YOLOv5x+ |
√ |
√ |
√ |
90.95 |
89.16 |
78.47 |
5 626 486 |
63.43 |
At the same time, for the problem that the detection time of the improved model is higher than that of the original model, we think it is worth it, because although the detection time of a single image has increased, the detection accuracy of the image has been greatly improved, and the detection time is milliseconds, so the impact in practical applications is not obvious.
- Why did the article choose CBAM attention module? What's the reason? If possible, please add other attention for comparative analysis, such as SENet, SAM, SKNet, etc;
Response: Thank you for your comment about details.As for why the attention module of CBAM is selected, this paper refers to the paper "WOO S, PARK J, LEE J, et al. CBAM: Collaborative block attention module [C] WOO S, PARK J, LEE J, et al. CBAM: Collaborative block attention module [C]//Proceedings of the European conference on computer vision (ECCV), March 19, 2018", in which the attention mechanism module is displayed, It is an attention mechanism module that combines spatial and channel. Compared with SENet's attention mechanism that only focuses on channels, it can achieve better results.
- Lack of comparative experiments with other mainstream classical networks;
Response: Thank you for your comment about details.Thank you for your guidance on our shortcomings. According to your suggestions, we added comparative experiments and discussed them. The results are as follows:
“4.5. Comparison of different network model training
In this study, ResNet-50, ResNet-18 and DesNet-201 networks are used to compare the performance of the improved models.
Table 3 shows that YOLOv5x+has the highest mAP value and the highest detection accuracy. Compared with ResNet-50, YOLOv5x+'s mAP is 7.34% higher. Compared with ResNet-18 and DesNet-201 models, the mAP of YOLOv5x+is increased by 7.89% and 2.89% respectively. In addition, the model proposed in this paper has significantly improved in precision, recall and detection time. Based on the comparison of the above experimental data, the improvement of this study is meaningful.
Table 3 Performance comparison of different models
|
P/% |
R/% |
Map/% |
Detection time/ms |
ResNet-50 |
80.10 |
71.67 |
71.14 |
74.86 |
ResNet-18 |
79.13 |
67.14 |
70.58 |
80.16 |
DesNet-201 |
84.30 |
74.95 |
75.58 |
69.73 |
YOLOv5x+ |
90.95 |
89.16 |
78.47 |
63.43 |
”(Line 415-425)
- It is not obvious that the detection accuracy is significantly improved in Figure 9, please add the detection effect drawing;
Response: Thanks for your careful review.According to your suggestion, we re detected the image and added a new detection effect to the article.
(a)YOLOv5x+ model (b)YOLOv5x model
Fig. 11 Comparison of test results
- Inappropriate language on lines 356-359. For example, how did the conclusion that "it is more suitable for the detection of cotton flowers in this study" come to?
Response: Thank you for your comment about details.We apologize for the inaccuracy in the text. We replace the words "it is more suitable for the detection of cotton flowers in this study" with“Although the detection time of a single image is increased, it is more suitable for cotton detection tasks than the original model.”
- Conclusion and Abstract sections are too similar;
Response: Thank you for your comment about details. According to your suggestion, we have revised the abstract and conclusion, and improved them as follows:
“Abstract: In order to study the detection effect of cotton boll opening after spraying defoliant, and to solve the problem of low efficiency of traditional manual detection methods for the use effect of cotton defoliant, this study proposed a cotton detection method improved YOLOv5x+algorithm. Convolution Attention Module (CBAM) is embedded after Conv to enhance the network's feature extraction ability, suppress background information interference, and enable the network to better focus on cotton targets in the detection process; At the same time, the depth separable convolution (DWConv) is used to replace the ordinary convolution (Conv) in YOLOv5x model, reducing the convolution kernel parameters in the algorithm, reducing the amount of calculation, and improving the detection speed of the algorithm; Finally, the detection layer is added to make the algorithm have higher accuracy in detecting small size cotton. The test results show that the accuracy rate P (%), recall rate R (%) and mAP value (%) of the improved algorithm reach 90.95, 89.16 and 78.47 respectively, which are 8.58, 8.84 and 5.15 higher than YOLOv5x algorithm respectively, and the convergence speed is faster, the error is smaller, and the resolution of cotton background and small target cotton is improved, which can meet the detection of cotton boll opening effect after spraying defoliant.”
“Conclusions:In this study, in view of the low efficiency of traditional cotton defoliant use effect detection and the low detection effect of cotton boll opening effect, based on the original YOLOv5x model, a depth learning model YOLOv5x+for detecting the effect of cotton defoliant was designed by embedding convolution attention module, using depth separable convolution, and adding detection layers for small size cotton. It can be seen from the test results that the average detection time of each image of YOLOv5x+model is 78.43ms, the accuracy rate P (%) and recall rate R (%) are 90.95 and 89.16, which are 8.58 and 8.84 percentage points higher than the original YOLOv5x model respectively, meeting the cotton detection task and having higher accuracy, so it is applicable to the cotton detection field. However, there are still cases of missing or error detection. The next part of the research will need to further optimize the attention mechanism of the model, improve the detection accuracy of the model, and deploy the improved model on the UAV to achieve real-time cotton detection.
”
- Some citations are in the wrong format.
Response: Thank you for your comment about details.I apologize for our mistakes. Now we have modified the reference format in the full text. Some improvements are as follows;
“Cotton was one of the most important crops in the world. It is also an important industrial raw material and strategic material. It plays an important role in the economic development of China and the world[1].As early cotton harvesting mode occupied a large number of human resources, the production cost per unit area of cotton was relatively high, which restricted the market competitiveness[2].In recent years, with the rapid development and promotion of mechanized harvesting technology of cotton, the advantages of large-scale, standardized and mechanized production of machine picked cotton are prominent, and the economic benefits are very significant[3]. Chemical defoliation and ripening is an important link of mechanical cotton picking and an important measure to promote the maturity of late maturing cotton[4]. The defoliating and ripening effect of the cotton defoliant can not only promote the natural cracking of cotton bolls and concentrated boll opening , but also effectively reduce the impurity content of mechanically harvested cotton, which directly affects the quality and efficiency of mechanically harvested cotton[5,6].”
Lv Xin, Liang Bin, Zhang Lifu, Ma Fuyu, Wang Haijiang, Liu Yangchun, Gao Pan, Zhang Ze, Hou Tongyu. Construction of an Agricultural Big Data Plat-form for XPCC Cotton Production[J]. Journal of Agricultural Big Data, 2020, 2 (01):70-78. doi:10.19788/j.issn.2096-6369.200109.
Mao Shuchun, Li Yabing, Wang Zhanbiao, Lei Yaping, Huang Qun, Wang Wenkui, Yang Beibei, Feng Lu, Li Pengcheng. Transformation and Upgrading of China's Cotton Industry under the Background of Agricultural High-quality Development[J]. Agricultural Outlook, 2018, 14(05 ): 39-45.
Chen Minzhi, Yang Yanlong, Wang Yuxuan, Tian Jingshan, Xu Shouzhen, Liu Ningning, Dangke, Zhang Wangfeng. Plant Type Characteristics and Evolution of Main Economic Characters in Early Maturing Upland Cotton Cultivar Replacement in Xinjiang[J]. Scientia Agricultura Sinica , 2019, 52 (19): 3279-3290. doi: 10.3864/j.issn.0578-1752.2019.19.001.
Liu Chan. Effects of different defoliants and their influence on cotton yield and quality [D]. Tarim University, 2021. doi: 10.27708/d.cnki.gtlmd.2021.000068.
Zhou Xianlin, Qin Qin, Wang Long, Li Lu, Hu Chengcheng, Hong Xiuchun, Wang Wei, Zhu Haiyong. Influence of Defoliant on Defoliation Effect and Fiber Quality of Cotton under Two Kinds of Mechanical Harvesting Modes[J]. Journal of Agricultural Science and Technology, 2020, 22(11):144-152. doi:10.13304/j.nykjdb.2019.0628.
Ma Qi, Li Jilian, Ning Xinzhu, Liu Ping, Deng Fujun, Lin Hai. Analysis on the of chemical defoliation and ripening of Xinluzao 60 under two kinds of mechanical cotton-picking planting modes [J].Journal of Chinese Agricultural Machinery, 2020, 41(05):139-144. doi:10.13733/j.jcam.issn.2095-5553.2020.05.023.
Best regards,
Can Hu

Reviewer 3 Report
1. The abstract is not clear, please highlight the innovative points of the manuscript;
2. In lines 74-75 and 245-247, English needs to be improved;
4. Figure 3 is not clear enough, and the marked details are not displayed enough;
5. In line 229 of the manuscript, why is the feature map output at the 17th, 20th and 23rd layers processed, what is the idea? What is the size of the feature maps output by layers 17, 20 and 23?
6. Figure 7 is not pretty. Please use different color squares to represent different types of convolution and modules;
7. In line 260-261, the classification threshold has a great influence on the experimental results. How is the threshold obtained? Why is there no analysis? Please add this part of the experiment to determine the optimal threshold;
8. Several parameters that must be important in algorithm improvement,, such as model size, model parameters, model FLOPs, AUC index, etc., are not reflected in Table 1. And Table 1 shows that the detection time of the improved model is 50% higher than that of the original model. Is it worth it?
10. Why did the article choose CBAM attention module? What's the reason? If possible, please add other attention for comparative analysis, such as SENet, SAM, SKNet, etc;
11. Lack of comparative experiments with other mainstream classical networks;
12. It is not obvious that the detection accuracy is significantly improved in Figure 9, please add the detection effect drawing;
13. Inappropriate language on lines 356-359. For example, how did the conclusion that "it is more suitable for the detection of cotton flowers in this study" come to?
14. Conclusion and Abstract sections are too similar;
15. Some citations are in the wrong format.
Author Response
Dear reviewer,
Thank you very much for giving us an opportunity to revise our manuscript entitled "Study on the detection of defoliation effect of an improved YOLOv5x cotton"(Manuscript ID:agriculture-1926388). We have studied the comments carefully. Those comments are valuable and helpful for revising and improving our paper, and they provide important guiding significance to our researchers.
We have studied comments carefully and have made correction which we hope meet with approval. Revised portion are marked in red in the paper.. In addition to the suggestions and problems put forward by experts, we have also modified and identified other problems found during the review.
The main correction in the paper and the responds to reviewer’s comments are as following:
Reviewer: 3
Comments and Suggestions for Authors:
- The abstract is not clear, please highlight the innovative points of the manuscript;
Response: Thanks for your careful review.Sorry for the unclear expression of our abstract.According to your suggestion, we have modified the abstract, which is as follows:
“In order to study the detection effect of cotton boll opening after spraying defoliant, and to solve the problem of low efficiency of traditional manual detection methods for the use effect of cotton defoliant, this study proposed a cotton detection method improved YOLOv5x+algorithm. Convolution Attention Module (CBAM) is embedded after Conv to enhance the network's feature extraction ability, suppress background information interference, and enable the network to better focus on cotton targets in the detection process; At the same time, the depth separable convolution (DWConv) is used to replace the ordinary convolution (Conv) in YOLOv5x model, reducing the convolution kernel parameters in the algorithm, reducing the amount of calculation, and improving the detection speed of the algorithm; Finally, the detection layer is added to make the algorithm have higher accuracy in detecting small size cotton. The test results show that the accuracy rate P (%), recall rate R (%) and mAP value (%) of the improved algorithm reach 90.95, 89.16 and 78.47 respectively, which are 8.58, 8.84 and 5.15 higher than YOLOv5x algorithm respectively, and the convergence speed is faster, the error is smaller, and the resolution of cotton background and small target cotton is improved, which can meet the detection of cotton boll opening effect after spraying defoliant.”(Line 15-29)
- In lines 74-75 and 245-247, English needs to be improved;
Response: Thanks for your careful review.We have revised our manuscript based on your comments.Substituting “These research cases show that YOLO algorithm has great potential in cotton detection.”(Line 76-77) with “These cases show that YOLO algorithm has great application significance in cotton detection.”(Line 76-77). Substituting “In the original YOLOv5x model, the feature maps output from the 17th, 20th and 23rd layers are used to detect small, medium and large targets respectively. However, due to the small area of small-sized cotton in the data picture, the target will be lost after multiple convolutions, resulting in missed detection.”(Line 261-265) with “In the original YOLOv5x model, the feature maps output from layers 17, 20, and 31 were used for small, medium, and large target detection, respectively, but because the small size cotton occupies a very small area in the data picture, the targets will be lost after multiple convolutions, resulting in missed detections. ”(Line 261-265)
- Figure 3 is not clear enough, and the marked details are not displayed enough;
Response: Thanks for your careful review.We apologize for the mistake in the text.Now we've made the following modifications to Figure 3
Fig. 3 Image annotation
- In line 229 of the manuscript, why is the feature map output at the 17th, 20th and 23rd layers processed, what is the idea? What is the size of the feature maps output by layers 17, 20 and 23?
Response: Thanks for your careful review.Why is the feature map output at the 17th, 20th and 23rd layers processed?Because the feature maps output from the 17th, 20th and 23rd layers are used to detect small, medium and large targets respectively, when the UAV takes a picture of cotton, due to the influence of flight height, the size of cotton in the picture is inconsistent, and the edge cotton and cotton covered by leaves account for a small proportion in the picture. In order to display cotton, the feature maps on the corresponding layers are processed.The output feature map sizes of layers 17, 20 and 23 are 80*80, 160*160, 40*40, respectively.
- Figure 7 is not pretty. Please use different color squares to represent different types of convolution and modules;
Response: Thanks for your suggestion.According to your suggestion, we used different colored squares to represent different types of convolutions and modules, and modified the text font size.
Fig. 7 Improved YOLOv5x network structure diagram
- In line 260-261, the classification threshold has a great influence on the experimental results. How is the threshold obtained? Why is there no analysis? Please add this part of the experiment to determine the optimal threshold;
Response: Thanks for your careful review.We are sorry for this mistake.According to your suggestion, we have added this part of experiment.The details are as follows:
“4.3.3. Determination of optimal threshold
The recognition model needs to filter the prediction box according to the preset threshold after predicting the confidence level of the target category. Based on the same recognition model, different confidence thresholds are used for prediction, and the accuracy and recall of the recognition results are different.
If the confidence threshold of the recognition model is not properly selected, the prediction results shown in Fig. 8 will appear: when the confidence threshold is set too low, cotton leaves in the image will be identified by mistake, and when the threshold is set too high, cotton targets may be missed. Therefore, it is necessary to determine appropriate confidence threshold for the model in combination with specific recognition tasks to accurately screen the cotton targets to be identified.
(a)Threshold is too low (b)Threshold is too high
Fig. 8 Impact of Threshold on Recognition Results
Based on the cotton recognition model obtained from the training, by adjusting the confidence threshold, compare the changes in the accuracy, recall and mAP of the measurement model for cotton target recognition of the test set under different thresholds, and determine the best prediction category threshold of the model in combination with the actual needs of cotton recognition tasks. Through test, the identification accuracy, recall and mAP distribution of the model under different confidence thresholds are shown in Fig. 9.
Fig.9 Changes of performance parameters of model with different confidence thresholds
”(Line 337-359)
- Several parameters that must be important in algorithm improvement,, such as model size, model parameters, model FLOPs, AUC index, etc., are not reflected in Table 1. And Table 1 shows that the detection time of the improved model is 50% higher than that of the original model. Is it worth it?
Response: Thanks for your careful review.The parameter problem in the algorithm improvement is not shown in Table 2, which is our fault. Thank you again for pointing out the shortcomings. Now we have added the parameter quantity index to the table, and the improvement is as follows:
Table 2 ablation test results
Model |
CBAM |
DWConv |
Small size cotton detection layer |
P/% |
R/% |
Map/% |
Parameter quantity |
Detection time/ms |
YOLOv5x |
× |
× |
× |
82.37 |
80.32 |
73.32 |
6 187 024 |
43.78 |
A |
√ |
× |
× |
84.59 |
82.96 |
72.43 |
4 082 034 |
49.21 |
B |
× |
√ |
× |
84.13 |
85.41 |
73.54 |
6 820 835 |
43.12 |
C |
× |
× |
√ |
86.64 |
84.40 |
75.19 |
6 759 216 |
65.76 |
YOLOv5x+ |
√ |
√ |
√ |
90.95 |
89.16 |
78.47 |
5 626 486 |
63.43 |
At the same time, for the problem that the detection time of the improved model is higher than that of the original model, we think it is worth it, because although the detection time of a single image has increased, the detection accuracy of the image has been greatly improved, and the detection time is milliseconds, so the impact in practical applications is not obvious.
- Why did the article choose CBAM attention module? What's the reason? If possible, please add other attention for comparative analysis, such as SENet, SAM, SKNet, etc;
Response: Thank you for your comment about details.As for why the attention module of CBAM is selected, this paper refers to the paper "WOO S, PARK J, LEE J, et al. CBAM: Collaborative block attention module [C] WOO S, PARK J, LEE J, et al. CBAM: Collaborative block attention module [C]//Proceedings of the European conference on computer vision (ECCV), March 19, 2018", in which the attention mechanism module is displayed, It is an attention mechanism module that combines spatial and channel. Compared with SENet's attention mechanism that only focuses on channels, it can achieve better results.
- Lack of comparative experiments with other mainstream classical networks;
Response: Thank you for your comment about details.Thank you for your guidance on our shortcomings. According to your suggestions, we added comparative experiments and discussed them. The results are as follows:
“4.5. Comparison of different network model training
In this study, ResNet-50, ResNet-18 and DesNet-201 networks are used to compare the performance of the improved models.
Table 3 shows that YOLOv5x+has the highest mAP value and the highest detection accuracy. Compared with ResNet-50, YOLOv5x+'s mAP is 7.34% higher. Compared with ResNet-18 and DesNet-201 models, the mAP of YOLOv5x+is increased by 7.89% and 2.89% respectively. In addition, the model proposed in this paper has significantly improved in precision, recall and detection time. Based on the comparison of the above experimental data, the improvement of this study is meaningful.
Table 3 Performance comparison of different models
|
P/% |
R/% |
Map/% |
Detection time/ms |
ResNet-50 |
80.10 |
71.67 |
71.14 |
74.86 |
ResNet-18 |
79.13 |
67.14 |
70.58 |
80.16 |
DesNet-201 |
84.30 |
74.95 |
75.58 |
69.73 |
YOLOv5x+ |
90.95 |
89.16 |
78.47 |
63.43 |
”(Line 415-425)
- It is not obvious that the detection accuracy is significantly improved in Figure 9, please add the detection effect drawing;
Response: Thanks for your careful review.According to your suggestion, we re detected the image and added a new detection effect to the article.
(a)YOLOv5x+ model (b)YOLOv5x model
Fig. 11 Comparison of test results
- Inappropriate language on lines 356-359. For example, how did the conclusion that "it is more suitable for the detection of cotton flowers in this study" come to?
Response: Thank you for your comment about details.We apologize for the inaccuracy in the text. We replace the words "it is more suitable for the detection of cotton flowers in this study" with“Although the detection time of a single image is increased, it is more suitable for cotton detection tasks than the original model.”
- Conclusion and Abstract sections are too similar;
Response: Thank you for your comment about details. According to your suggestion, we have revised the abstract and conclusion, and improved them as follows:
“Abstract: In order to study the detection effect of cotton boll opening after spraying defoliant, and to solve the problem of low efficiency of traditional manual detection methods for the use effect of cotton defoliant, this study proposed a cotton detection method improved YOLOv5x+algorithm. Convolution Attention Module (CBAM) is embedded after Conv to enhance the network's feature extraction ability, suppress background information interference, and enable the network to better focus on cotton targets in the detection process; At the same time, the depth separable convolution (DWConv) is used to replace the ordinary convolution (Conv) in YOLOv5x model, reducing the convolution kernel parameters in the algorithm, reducing the amount of calculation, and improving the detection speed of the algorithm; Finally, the detection layer is added to make the algorithm have higher accuracy in detecting small size cotton. The test results show that the accuracy rate P (%), recall rate R (%) and mAP value (%) of the improved algorithm reach 90.95, 89.16 and 78.47 respectively, which are 8.58, 8.84 and 5.15 higher than YOLOv5x algorithm respectively, and the convergence speed is faster, the error is smaller, and the resolution of cotton background and small target cotton is improved, which can meet the detection of cotton boll opening effect after spraying defoliant.”
“Conclusions:In this study, in view of the low efficiency of traditional cotton defoliant use effect detection and the low detection effect of cotton boll opening effect, based on the original YOLOv5x model, a depth learning model YOLOv5x+for detecting the effect of cotton defoliant was designed by embedding convolution attention module, using depth separable convolution, and adding detection layers for small size cotton. It can be seen from the test results that the average detection time of each image of YOLOv5x+model is 78.43ms, the accuracy rate P (%) and recall rate R (%) are 90.95 and 89.16, which are 8.58 and 8.84 percentage points higher than the original YOLOv5x model respectively, meeting the cotton detection task and having higher accuracy, so it is applicable to the cotton detection field. However, there are still cases of missing or error detection. The next part of the research will need to further optimize the attention mechanism of the model, improve the detection accuracy of the model, and deploy the improved model on the UAV to achieve real-time cotton detection.
”
- Some citations are in the wrong format.
Response: Thank you for your comment about details.I apologize for our mistakes. Now we have modified the reference format in the full text. Some improvements are as follows;
“Cotton was one of the most important crops in the world. It is also an important industrial raw material and strategic material. It plays an important role in the economic development of China and the world[1].As early cotton harvesting mode occupied a large number of human resources, the production cost per unit area of cotton was relatively high, which restricted the market competitiveness[2].In recent years, with the rapid development and promotion of mechanized harvesting technology of cotton, the advantages of large-scale, standardized and mechanized production of machine picked cotton are prominent, and the economic benefits are very significant[3]. Chemical defoliation and ripening is an important link of mechanical cotton picking and an important measure to promote the maturity of late maturing cotton[4]. The defoliating and ripening effect of the cotton defoliant can not only promote the natural cracking of cotton bolls and concentrated boll opening , but also effectively reduce the impurity content of mechanically harvested cotton, which directly affects the quality and efficiency of mechanically harvested cotton[5,6].”
Lv Xin, Liang Bin, Zhang Lifu, Ma Fuyu, Wang Haijiang, Liu Yangchun, Gao Pan, Zhang Ze, Hou Tongyu. Construction of an Agricultural Big Data Plat-form for XPCC Cotton Production[J]. Journal of Agricultural Big Data, 2020, 2 (01):70-78. doi:10.19788/j.issn.2096-6369.200109.
Mao Shuchun, Li Yabing, Wang Zhanbiao, Lei Yaping, Huang Qun, Wang Wenkui, Yang Beibei, Feng Lu, Li Pengcheng. Transformation and Upgrading of China's Cotton Industry under the Background of Agricultural High-quality Development[J]. Agricultural Outlook, 2018, 14(05 ): 39-45.
Chen Minzhi, Yang Yanlong, Wang Yuxuan, Tian Jingshan, Xu Shouzhen, Liu Ningning, Dangke, Zhang Wangfeng. Plant Type Characteristics and Evolution of Main Economic Characters in Early Maturing Upland Cotton Cultivar Replacement in Xinjiang[J]. Scientia Agricultura Sinica , 2019, 52 (19): 3279-3290. doi: 10.3864/j.issn.0578-1752.2019.19.001.
Liu Chan. Effects of different defoliants and their influence on cotton yield and quality [D]. Tarim University, 2021. doi: 10.27708/d.cnki.gtlmd.2021.000068.
Zhou Xianlin, Qin Qin, Wang Long, Li Lu, Hu Chengcheng, Hong Xiuchun, Wang Wei, Zhu Haiyong. Influence of Defoliant on Defoliation Effect and Fiber Quality of Cotton under Two Kinds of Mechanical Harvesting Modes[J]. Journal of Agricultural Science and Technology, 2020, 22(11):144-152. doi:10.13304/j.nykjdb.2019.0628.
Ma Qi, Li Jilian, Ning Xinzhu, Liu Ping, Deng Fujun, Lin Hai. Analysis on the of chemical defoliation and ripening of Xinluzao 60 under two kinds of mechanical cotton-picking planting modes [J].Journal of Chinese Agricultural Machinery, 2020, 41(05):139-144. doi:10.13733/j.jcam.issn.2095-5553.2020.05.023.
